# Effectiveness of Volatile Natural Deep Eutectic Solvents (VNADESs) for the Green Extraction of *Chelidonium majus* Isoquinoline Alkaloids

**DOI:** 10.3390/molecules27092815

**Published:** 2022-04-28

**Authors:** Maciej Strzemski, Sławomir Dresler, Beata Podkościelna, Kamil Skic, Ireneusz Sowa, Daniel Załuski, Rob Verpoorte, Sylwia Zielińska, Paweł Krawczyk, Magdalena Wójciak

**Affiliations:** 1Department of Analytical Chemistry, Medical University of Lublin, 20-093 Lublin, Poland; slawomir.dresler@umlub.pl (S.D.); i.sowa@umlub.pl (I.S.); 2Department of Plant Physiology and Biophysics, Institute of Biological Science, Maria Curie-Skłodowska University, 20-033 Lublin, Poland; 3Faculty of Chemistry, Institute of Chemistry, Maria Curie-Skłodowska University, 20-031 Lublin, Poland; beatapod@poczta.umcs.lublin.pl; 4Institute of Agrophysics, Polish Academy of Sciences, 20-290 Lublin, Poland; k.skic@ipan.lublin.pl; 5Department of Pharmaceutical Botany and Pharmacognosy, Ludwik Rydygier Collegium Medicum, Nicolaus Copernicus University, 85-094 Bydgoszcz, Poland; daniel.zaluski@cm.umk.pl; 6Natural Products Laboratory, Institute of Biology, Leiden University, 2300RA Leiden, The Netherlands; verpoort@chem.leidenuniv.nl; 7Department of Pharmaceutical Biology and Biotechnology, Division of Pharmaceutical Biotechnology, Wroclaw Medical University, 50-556 Wroclaw, Poland; sylwia.zielinska@umed.wroc.pl; 8Immunology and Genetics Laboratory, Department of Pneumonology, Oncology and Allergology, Medical University of Lublin, 20-093 Lublin, Poland; krapa@poczta.onet.pl

**Keywords:** green chemistry, NADESs, isoquinoline alkaloids, protopine, chelidonine, berberine, chelerythrine, coptisine, sanguinarine, greater celandine

## Abstract

The *Chelidonium majus* plant is rich in biologically active isoquinoline alkaloids. These alkaline polar compounds are isolated from raw materials with the use of acidified water or methanol; next, after alkalisation of the extract, they are extracted using chloroform or dichloromethane. This procedure requires the use of toxic solvents. The present study assessed the possibility of using volatile natural deep eutectic solvents (VNADESs) for the efficient and environmentally friendly extraction of *Chelidonium* alkaloids. The roots and herb of the plant were subjected three times to extraction with various menthol, thymol, and camphor mixtures and with water and methanol (acidified and nonacidified). It has been shown that alkaloids can be efficiently isolated using menthol–camphor and menthol–thymol mixtures. In comparison with the extraction with acidified methanol, the use of appropriate VNADESs formulations yielded higher amounts of protopine (by 16%), chelidonine (35%), berberine (76%), chelerythrine (12%), and coptisine (180%). Sanguinarine extraction efficiency was at the same level. Additionally, the values of the contact angles of the raw materials treated with the tested solvents were assessed, and higher wetting dynamics were observed in the case of VNADESs when compared with water. These results suggest that VNADESs can be used for the efficient and environmentally friendly extraction of *Chelidonium* alkaloids.

## 1. Introduction

The greater celandine (*Chelidonium majus* L., Papaveraceae) is a medicinal plant used since antiquity. In folk medicine, it has mainly been used in the treatment of eye, skin, and liver diseases and parasitic infections. The popularity of the use of this species is also associated with its wide occurrence and easy access to the raw material. The current pharmacological studies of *C. majus* extracts are mainly focused on the assessment of their antibacterial, antiviral, anti-inflammatory, and immunomodulating activity. Research on their antiproliferative and proapoptotic potential is underway as well. Concurrently, there are investigations aimed at the detection of pure chemical compounds and their complexes responsible for the pharmacological activity of extracts from this plant. To date, berberine (Berb), chelerythrine (Chele), chelidonine (Che), coptisine (Cop), protopine (Prot), and sanguinarine (Sang) have been shown to be the main pharmacologically active metabolites of *C. majus* [1,2,3]. The structures of these compounds are shown in Figure 1. These compounds are present as salts and free bases. They are isolated from raw material through extraction usually carried out with acidified water (conversion into water-soluble salts). Next, the extract is alkalised, and the bases are subjected to liquid–liquid extraction with the use of dichloromethane, butanol, or chloroform [4,5,6,7,8]. The procedure of isolation of *C. majus* alkaloids is therefore a multistep process requiring the use of toxic organic solvents. The use of natural deep eutectic solvents (NADESs), i.e., mixtures of natural substances characterised by a substantially lower melting point than their individual components, may limit the amount of toxic chemical compounds necessary at the stage of isolation of plant metabolites [9]. These mixtures have been successfully used for the extraction of polyphenolic compounds [10], e.g., flavonoids [11,12,13], ginsenosides and chlorogenic acid [13], xanthones [14], anthocyanins [9,15], and alkaloids [16,17]. However, NADESs have some drawbacks that hinder their practical application. The high viscosity of most NADESs impedes the process of mixing the extract with the plant matrix, separating the matrix from the extract after completion of the extraction [10], and analysing the extracts with the use of modern instrumental analysis techniques [13]. Due to the negligible volatility of NADESs, the acquisition of dry extract residues through evaporation of the extractant is impossible. These problems were partially solved by applying a menthol–thymol mixture for the extraction of pentacyclic triterpenes, where the NADES volatile components facilitate complete evaporation of the extractant [18].

The research on the use of NADESs conducted to date has mainly focused on the extraction of polar acid metabolites [9,12,13]. An exception is triterpene acids, as they were isolated as nonpolar compounds with the use of NADESs, which are mixtures of nonpolar substances [18]. To the best of our knowledge, there are no reports on the use of volatile hydrophobic NADESs for the isolation of alkaline polar compounds, such as *C. majus* isoquinoline alkaloids (IA), which may be good model compounds for the assessment of the lipophilic NADES-based extraction potential.

This report provides an answer to the question of the possibility of using volatile hydrophobic NADESs (VNADESs) for the efficient extraction of the IA directly from the roots and herb of *C. majus*, without the necessity of employing a two-step extraction procedure and heavy chemistry. Such a solution may facilitate an economical and efficient industrial isolation of IA with a simultaneous reduction of the negative impact of industry on the natural environment.

## 2. Results and Discussion

### 2.1. Extraction Efficiency

The present experiment demonstrated the possibility of the efficient extraction of *C. majus* IA with the use of VNADESs. It has been evidenced that it is possible to use VNADESs for the successful isolation of substantial amounts of IA, especially from *C. majus* roots, which are a much richer raw material than the herb [1]. As shown by the literature data, methanol (MeOH), ethanol, or water most often with the addition of hydrochloric or acetic acid have been used most frequently for the isolation of these compounds to date [19,20,21,22]. Hence, pure and acidified (0.05 M HCl) water and pure and acidified (0.05 M HCl) MeOH were used as the control extractants in this study. The concentrations of most of the isolated analytes in the extracts were shown to increase in the following order: MeOH_HCl_ > MeOH > H_2_O_HC l_> H_2_O (Figure 2). Acidified MeOH, which exhibited the highest efficiency of isolation of Prot (0.49), Che (2.80), Berb (0.33), Chele (2.31), Cop (1.07), and Sang (1.71 mg∙g^−1^ DW) turned out to be the best extractant. The values determined in extracts obtained with the use of this solvent were regarded as 100% and were compared with those determined in the VNADES extracts. It was found that the menthol–thymol (MT) and thymol–camphor (TC) mixtures were good extractants of *C. majus* IA, whereas the menthol–camphor (MC) mixtures did not yield satisfactory extraction results. The MT 5:5, TC 5:5, and TC 6:4 mixtures were the best extractants of IA from the roots. The use of MT 5:5, TC 5:5, and TC 6:4 yielded the following contents of analytes: Prot +16, +2, and +12%; Che +35, +21, and +28%; Berb+21, +76, and +6%; Chele +12, -21, and +4%; Cop +180, +156, and +150; and Sang −2, −17, and −12% versus MeOH_HCl_, respectively. Thus, the application of VNADESs, especially MT 5:5, contributed to a significant increase in the *C. majus* IA extraction yield. The total IA content was 12.03 ± 0.45 mg∙g^−1^ DW in the extracts obtained with the use of MT 5:5 and 8.7 ± 0.37 mg∙g^−1^ DW in the MeOH_HCl_ variants (25% less than in MT 5:5). These results reveal an incomplete extraction of IA with the most commonly used solvents. The improved efficiency of the IA extraction using VNADESs can therefore indicate the possibility of economical, efficient, and environmentally friendly isolation of these compounds on an industrial scale. The use of mixtures of volatile compounds will facilitate obtaining dry residue extracts, which is an advantage over classically used NADESs.

Similarly, the MT 5:5 mixture ensured the most efficient extraction from the herb. The content of IA, except for Cop (2.50 ± 0.14 mg∙g^−1^ DW), was below 0.1 mg∙g^−1^ DW of the raw material, and no Sang and Chele were detected (Appendix A). It therefore seems that the herb of the analysed species has marginal importance as a raw material for potential industrial isolation of IA. The examples of HPLC–DAD chromatograms of the extracts obtained with VNADESs are shown in Appendix A.

### 2.2. Physical Properties of Extractants

The physical properties of extractants exert a significant impact on the extraction efficiency. Hence, the density and surface tension parameters of the obtained VNADESs were investigated. The results obtained are shown in Table 1.

Next, the values of the contact angles of the plant material subjected to VNADES-based extraction and treated with the control extractants were determined. The contact angle of 0° indicates complete infiltration of the solid surface by the liquid. The higher the angle value, the lower the infiltration degree [23]; therefore, lower wetting dynamics and a deterioration of the extraction efficiency should be expected. The following plant material wetting dynamics were shown in this study: MeOH and MeOH_HCl_ > MC > MT > TC > H_2_O and H_2_O_HCl_ (Appendix A). The MT and TC mixtures, which exhibited the highest efficiency in the IA extraction, wetted the extracted matrix more poorly than MC; therefore, there seems to be no direct relationship between the values of contact angles and the IA extraction yield. However, in the case of the TC mixtures, the increase in the thymol content versus the camphor content increased the wettability of the plant matrix, which may be associated with the increase in the IA extraction efficiency of mixtures with the higher thymol content. The finding that all the VNADESs infiltrated the plant matrix faster than water can be highly important for the technical aspects of the extraction process.

### 2.3. Principal Component Analysis

The PCA analysis showed that both Factor 1 and Factor 2 explained 80.37% and 89.35% of total variability for the leaves and roots, respectively (Figure 3). In the leaves, Factor 1 was negatively loaded by the contact angle, surface tension, and density of the extractants, while Factor 2 was negatively determined by IA. This showed a separation of three major groups of samples: the first one grouped the water extractant with higher surface tension and contact angle. The second group was located above the X-axis and separated the TC, MC, and MT 6:4 extractants with a low capacity of IA extraction. The third group (methanol and MT 5:5) showed a large affinity to IA extraction. In the case of the roots, a negative correlation of surface tension, contact angle, and density of extractants with the capacity of IA extraction along Factor 1 was noted. Additionally, most variables exhibited a negative correlation with Factor 2. Factor 1 separated the water, MC, and TC4:6 extractants with a relatively lower capacity of IA extraction than those with the higher capability of IA extraction (MeOH, MT, TC 5:5, 6:4).

## 3. Materials and Methods

### 3.1. Reference Standards and Chemicals

Alkaloid standards: hydrochloride of chelidonine and protopine, chloride of berberine, chelerythrine, coptisine, sanguinarine, thymol, DL-menthol, and (±)-camphor were purchased from Sigma-Aldrich (St. Louis, MO, USA). Ammonium acetate, hydrochloric acid, acetic acid, methanol, and HPLC-grade acetonitrile were purchased from Merck (Darmstadt, Germany). Water was deionised and purified by ULTRAPURE Milipore Direct-Q^®®^ 3UV–R (Merck).

### 3.2. Preparation of VNADESs and Reference Extractants

Two-component mixtures of menthol–thymol (MT), menthol–camphor (MC), and thymol–camphor (TC) were obtained by mixing the components in a specific mass:mass ratio at room temperature. The mixtures, which are liquids at room temperature, had the following quantitative compositions: MT 5:5 and 6:4, MC 6:4 and 7:3, and TC 4:6, 5:5, and 6:4 (*w*/*w*). Water and MeOH used as control extractants were acidified with hydrochloric acid to a concentration of 0.05 M.

### 3.3. Plant Material and Extraction Procedure

The *C. majus* plants were obtained from the Botanical Garden of Maria Curie-Skłodowska University in Lublin, Poland (51°16′ N, 22°30′ E). The plants were collected in the full flowering phase in May 2020. The plants were divided into the roots and shoots. The plant material was frozen at −80 °C and freeze-dried (0.001 mbar for 72 h) before extraction using a Christ Alpha 2-4 LDplus laboratory freeze dryer (Martin Christ Gefriertrocknungsanlagen GmbH, Osterode am Harz, Germany). The dry material was powdered using a laboratory grinder IKA A11 (IKA-Werke, Staufen, Germany) and sieved (1.6 mm sieve). The powder (0.05 g) was extracted three times with a fresh portion of VNADESs or control extractants—Acidified and nonacidified methanol and water (2 mL, 2 mL, and 1 mL) in an ultrasonic bath at a frequency of 35 kHz, (Sonorex RK 512 H, Bandelin, Berlin, Germany) for 15 min at ambient temperature. The extracts were centrifuged using a Sigma 1-16K laboratory microcentrifuge (Sigma Laborzentrifugen GmbH, Osterode am Harz, Germany) at 20,000 g for 10 min, combined, and filled up to 5 mL in a volumetric flask. The plant samples were deposited at the Department of Analytical Chemistry, Medical University of Lublin (voucher specimen no. 05.2020).

### 3.4. HPLC Analysis

IA were determined using the methodology published previously [24]. Data were recalculated per gram of dry weight (DW).

### 3.5. Determination of the Physical Properties of the Extractants

The contact angles and liquid surface tension measurements were performed using a DSA100 goniometer (Krüss, Germany). The contact angle was determined at the intersection of the drop contour line with the flat sample surface according to the procedure for sessile drop method described by Dang-Vu et al. [25]. The liquid surface tension was measured under constant temperature conditions (20 °C) using pendant drop method and the Young–Laplace fitting mode. Liquid density was determined after weighing the liquid volume on a high accuracy laboratory balance.

### 3.6. Statistical Analysis

Both the principal component analysis (PCA) and one-way analysis of variance with Tukey’s post hoc test were performed using Statistic ver. 13 software (TIBCO Software Inc. 2017, Palo Alto, CA, USA).

## 4. Conclusions

In conclusion, this study verifies that naturally occurring volatile compounds (especially a mixture of equal amounts of menthol and thymol) enhance the extraction of alkaloids from *C. majus*. Using these mixtures, chelidonium alkaloids can be isolated with much higher yields than with acidified methanol, which is currently one of the most commonly used extractants and does not meet the assumptions of “green extraction”. These findings confirm the potential of using VNADESs in the *C. majus* alkaloids extraction and are in the line with a green chemistry concept.

## Figures and Tables

**Figure 1 molecules-27-02815-f001:**
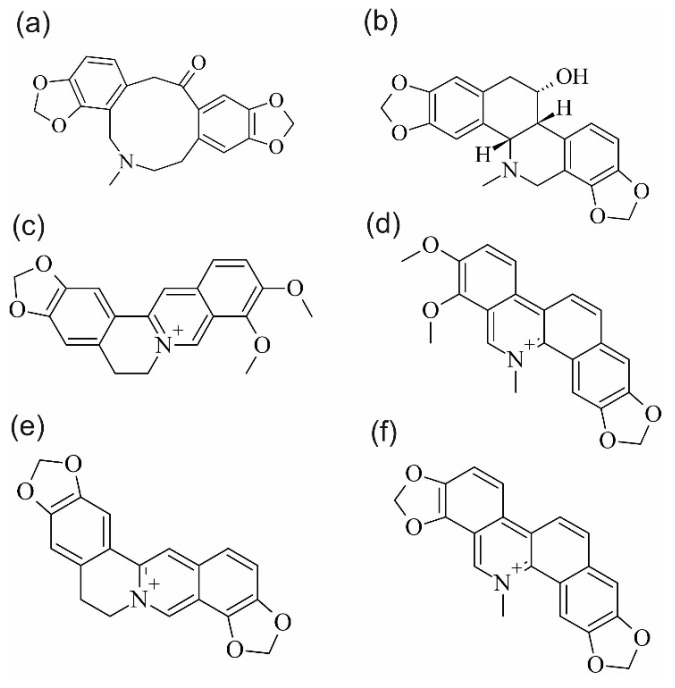
Chemical structures of major alkaloids in *Chelidonium majus* L.: (**a**)—Protopine, (**b**)—Chelidonine, (**c**)—Berberine, (**d**)—Chelerythrine, (**e**)—Coptisine, and (**f**)—Sanguinarine.

**Figure 2 molecules-27-02815-f002:**
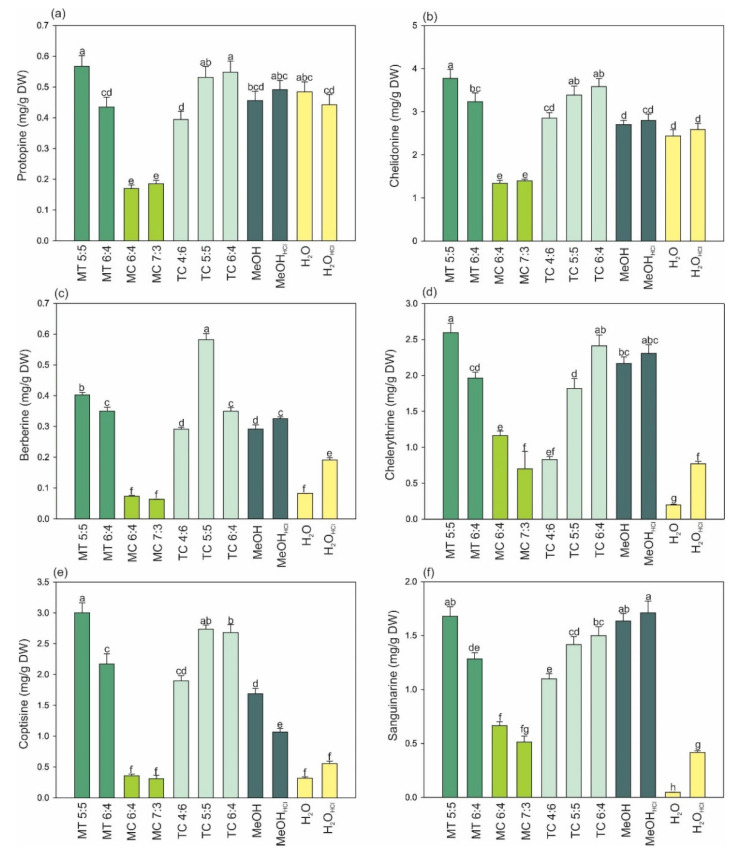
Yields of extraction of isoquinoline alkaloids from *Chelidonium majus* roots with volatile natural deep eutectic solvents and commonly used extractants; (**a**–**f**), yields of extraction: protopine, chelidonine, berberine, chelerythrine, coptisine, and sanguinarine respectively. MT—Menthol–thymol mixtures; MC—Menthol-camphor mixtures; TC—Thymol-camphor mixtures. The mass:mass ratio of the components in the mixtures is shown next to the symbols. MeOH_HCl_ and H_2_O_HCl_-MeOH and water acidified with hydrochloric acid to a concentration of 0.05 M. Data are mean ± SE (*n* = 5); values for individual raw materials followed by the same letter are not significantly different (*p* < 0.05, Tukey’s test).

**Figure 3 molecules-27-02815-f003:**
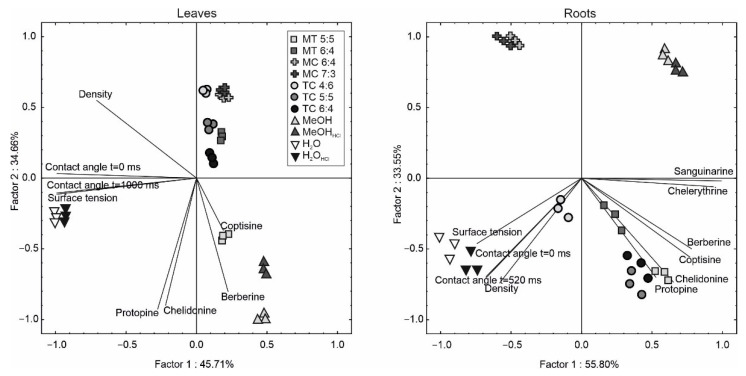
Scaled scatter plot of the principal component analysis of isoquinoline alkaloids and physical properties of extractants (MT—Menthol-thymol mixtures; MC—Menthol-camphor mixtures; TC—Thymol-camphor mixtures. The mass:mass ratio of the components in the mixtures is shown next to the symbols. MeOH_HCl_ and H_2_O_HCl_-MeOH and water acidified with hydrochloric acid to a concentration of 0.05 M. The length of the lines shows a correlation between original data and factor axes.

**Table 1 molecules-27-02815-t001:** Values of density and surface tension measured for the obtained VNADESs and control extractants.

	TC	MT	MC	MeOH	H_2_O
Density (g/cm^3^)	0.95	0.93	0.91	0.79	0.99
Surface tension (J/m^2^)	32	30	29	24	72

## Data Availability

Data is contained within the article or Appendix A.

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
