# Peer review of "Effectiveness of Volatile Natural Deep Eutectic Solvents (VNADESs) for the Green Extraction of Chelidonium majus Isoquinoline Alkaloids"

_molecules, 2022, doi:10.3390/molecules27092815_

Round 1
Reviewer 1 Report
The authors presented an efficient method for efficient separation of isoquinoline alkaloids (IA) using deep eutectic solvents (such as menthol-thymol mixture).
My suggestions for the authors are:
- Add the chemical structure of the isoquinoline alkaloids (IA) analysed.
- Add representative HPLC chromatograms of the alkaloids extracted with different VNADES solvents.
- Increase the size of the figures
- did the authors analyze the bioactivity of IA after extraction? Have they evaluated the antioxidant activity?
Author Response
REVIEWER’S RESPONSE:
Add the chemical structure of the isoquinoline alkaloids (IA) analysed.
AUTHORS’ RESPONSE: The structures of the analyzed compounds have been added as Figure 1.
REVIEWER’S RESPONSE:
Add representative HPLC chromatograms of the alkaloids extracted with different VNADES solvents.
AUTHORS’ RESPONSE: The chromatograms of the analyzed extracts have been added as Figure S2.
REVIEWER’S RESPONSE:
Increase the size of the figures
AUTHORS’ RESPONSE: This has been corrected.
REVIEWER’S RESPONSE:
Did the authors analyze the bioactivity of IA after extraction? Have they evaluated the antioxidant activity?
AUTHORS’ RESPONSE: At this stage, we did not test the activity of the obtained extracts. We wanted to obtain information on the extraction efficiency of compounds whose biological activity is already relatively well understood. However, we do not rule out conducting such research in the future.
Reviewer 2 Report
Dear editor:
I think this manuscuipt can be accecpted fater finishing the following work:
- Line 95, add a comma to the word “water”;
- Line 97, correct the “H2OHCl”;
- Line 99, “Prot 0.49, Che 2.80, Berb 0.33, Chele 2.31, Cop 1.07, and Sang 1.71 mg∙g-1 DW” is obscure;
Author Response
REVIEWER’S RESPONSE:
Line 95, add a comma to the word “water”;
AUTHORS’ RESPONSE: This has been corrected.
REVIEWER’S RESPONSE:
Line 97, correct the “H2OHCl”;
AUTHORS’ RESPONSE: This has been corrected.
REVIEWER’S RESPONSE:
Line 99, “Prot 0.49, Che 2.80, Berb 0.33, Chele 2.31, Cop 1.07, and Sang 1.71 mg∙g-1 DW” is obscure;
AUTHORS’ RESPONSE: This sentence has been replaced as follows: “Acidified MeOH, which exhibited the highest efficiency of isolation of Prot (0.49), Che (2.80), Berb (0.33), Chele (2.31), Cop (1.07), and Sang (1.71 mg∙g-1 DW), turned out to be the best extractant. The values determined in extracts obtained with the use of this solvent were regarded as 100% and were compared with those determined in VNADES extracts.”
Reviewer 3 Report
The authors performed an interesting initial study using a green method with volatile natural deep-eutectic solvents to extract alkaloids from roots and aereal parts of Chelidonium majus. The simplicity of the paper justifies it presentation as communication. However, despite being well written, the paper needs some corrections. My most important comments are regarding the plant organs extracted and the use of PCA. For the plant organs, if the authors have already known that the aereal parts had low content of alkaloids they could focus their paper only in the roots (justifying this in the introduction). Regarding the PCA, the authors should make clearer why presenting this result is important to the reader, since in the "results and discussion" section, they just described the results, without explaining the importance of the clusterings found. Additionally, the text need some datailing (especially in M&M) and some corrections as well (please check the attached PDF file), and the conclusion must be re-written. Based on the mentioned above, I reccomend minor review for the manuscript.

Author Response
REVIEWER’S RESPONSE:
The authors performed an interesting initial study using a green method with volatile natural deep-eutectic solvents to extract alkaloids from roots and aereal parts of Chelidonium majus. The simplicity of the paper justifies it presentation as communication. However, despite being well written, the paper needs some corrections. My most important comments are regarding the plant organs extracted and the use of PCA. For the plant organs, if the authors have already known that the aereal parts had low content of alkaloids they could focus their paper only in the roots (justifying this in the introduction). Regarding the PCA, the authors should make clearer why presenting this result is important to the reader, since in the "results and discussion" section, they just described the results, without explaining the importance of the clusterings found. Additionally, the text need some datailing (especially in M&M) and some corrections as well (please check the attached PDF file), and the conclusion must be re-written. Based on the mentioned above, I reccomend minor review for the manuscript.
AUTHORS’ RESPONSE:
We agree with the reviewer that the literature provides ample evidence of higher alkaloid content in the root than in the Chelidonium herb. However, we did not assume this a priori and wanted to evaluate the extraction efficiency from the herb as well. This allowed us to address the issue of alkaloid isolation from Chelidonium in a comprehensive manner using our model.
In the Results and discussion section, all obtained data and relationships between NADES and alkaloid extraction were described in detail. As we realized that raw data are not clear in not deep reading, we have tried to make it clearer. In this context, the PCA for both organs were performed; however, this analysis did not bring any new findings or conclusions but rather makes the data more easily readable.
Changes made based on the attached pdf document:
Abstract:
REVIEWER’S RESPONSE:
Based on what was presented, maybe the authors can focuse only in roots. What do the authors mean by herb? aereal parts?
AUTHORS’ RESPONSE: As we mentioned above, the herb extraction studies were designed to comprehensively address the issue of VNADES application. Whole flowering aerial parts were used in the study and this raw material was called herb.
REVIEWER’S RESPONSE:
Is there an specific proportion of them in the mixture?
AUTHORS’ RESPONSE: Of course, the extraction efficiency depends on the proportions in which the VNADES components are used, but here, because of the limited number of words (max 200), we were forced to limit the details. The manuscript is very short and the reader will very quickly find out what proportions are most beneficial.
Introduction:
REVIEWER’S RESPONSE:
Are these references for all the first part of the introduction? Please, put the references in the sentences above, avoiding mentioning them only here.
AUTHORS’ RESPONSE: Dear Reviewer, it seems to us that this information is quite basic and very general. It can be found not only in the three cited papers but also in many other manuscripts presenting the results of pharmacological and phytochemical studies of Chelidonium. In addition, we wanted to keep the manuscript as readable and simple as possible and for this reason we have taken the liberty of maintaining the current way of citation.
REVIEWER’S RESPONSE:
line 69 - Ref.
AUTHORS’ RESPONSE: References have been added.
REVIEWER’S RESPONSE: line 83 - of employing
AUTHORS’ RESPONSE: This has been corrected.
Results and Discussion:
REVIEWER’S RESPONSE:
Actually this sequence is not so clear. According to the results in the fig1, some of these "control" treatments were statistically equal. For others, MeOH without acid was the best. Only for berberine the acidified MeOH was the best solvent (not statistically similar to any other "control" extraction). I suggest that the authors rephrase this part to justify why do they chose the MeOH-HCl as standard procedure.
AUTHORS’ RESPONSE: In fact, it may seem that in many cases pure methanol is a better extractant than acidified methanol. However, we draw your attention to the fact that in no case (except for coptisine) were these differences not significant. Taking into account the efficiency of coptisine isolation, the choice of acidified methanol seemed to be more appropriate than the choice of non-acidified methanol.
REVIEWER’S RESPONSE:
Line 99: are these values the amounts? It is not clear, presenting in this way.
AUTHORS’ RESPONSE: This sentence has been modified according to the Reviewer’s 2 suggestion.
REVIEWER’S RESPONSE:
Lines 105-110: The authors do not need to write the contents in mg/g since the values are already present in the graphs. They should rephrase this part, and let only the percentage of increasing compared to the control.
AUTHORS’ RESPONSE: This has been corrected according to the Reviewer’s suggestion.
REVIEWER’S RESPONSE:
Lines 112-113: Actually, they were statistically similar, according to the results presented.
AUTHORS’ RESPONSE: We would like to thank the Reviewer for this pertinent remark. The sentence "except for sanguinarine, which was more efficiently extracted using MeOHHCl" has been deleted.
REVIEWER’S RESPONSE:
Lines 113-114: please include SD
AUTHORS’ RESPONSE: Exact values ± SD are given.
REVIEWER’S RESPONSE:
The authors could discuss a bit more the green extraction process, the use of NADES and especially VNADES in extration of alkaloids. The literature speaks few about this. But the authors should discuss why they decide to conduct the extraction of alkaloids with menthol, thymol and camphor.
AUTHORS’ RESPONSE: We have added a sentence about the advantage of VNADES over the previously used NADES.
Line 120 - This has been corrected.
Lines 127-130: If the authors already know the insignificant content of IA in the aereal parts, maybe they could remove this analysis from the communication and focus only on the roots, justifying in the introduction the focus on this plant organ.
AUTHORS’ RESPONSE: As mentioned, we wanted a comprehensive coverage of this study. It seems to us that two sentences do not negatively influence the volume of a short communication and make it more complete. If the reviewer allows, we would like to leave this short information.
REVIEWER’S RESPONSE:
Lines 135-137: These results could be presented in a table.
AUTHORS’ RESPONSE: Following the Reviewer's recommendations, we have presented the data in the form of a table.
3.2. Preparation of VNADES and reference extractants
Lines 183-187 have been modified according to the Reviewer's recommendations.
3.3. Plant material and extraction procedure
REVIEWER’S RESPONSE:
Line 192: comprasing stem, leaves and flowers?
AUTHORS’ RESPONSE: Yes, the whole flowering shoots were the herbal material used in herbal medicine (Herba Chelidoinii).
Lines 198 and 202-203 Has been corrected according to the Reviewer’s suggestion.
3.4. HPLC analysis
REVIEWER’S RESPONSE:
Please describe at least briefly the analysis. Is this a MS detector? A DAD detector? Did the authors use standards to construct a curve?
AUTHORS’ RESPONSE: Dear Reviewer, we did not want to over-build the manuscript, which is a short communication. We have added the methodological details of the HPLC analysis in the supplementary materials under the examples of chromatograms (Fig. S2).
4. Conclusions
The conclusions has been corrected according to the Reviewer’s sugges